# Adolescent's Collective Intelligence: Empirical Evidence in Real and Online Classmates Groups

**Enrico Imbimbo** [1],*,† , **Federica Stefanelli** [1],*,† **and Andrea Guazzini** [2],*

[1]  Department of Education, Languages, Intercultures, Literatures and Psychology, University of Florence, Via di San Salvi, 12, Building 26, 50135 Florence, Italy

[2]  Center for Study of Complex Dynamics (C.S.D.C), University of Florence, Via Sansone, 1, 50019 Sesto Fiorentino, Italy

*  Correspondence: enrico.imbimbo@unifi.it (E.I.); federica.stefanelli@unifi.it (F.S.); andrea.guazzini@unifi.it (A.G.)

†  These authors contributed equally to this work.

**Abstract:** Humans create teams to be more successful in a large variety of tasks. Groups are characterized by an emergent property called collective intelligence, which leads them to be smarter than single individuals. Previous studies proved that collective intelligence characterizes both real and online environments, focusing on adults' performances. In this work, we explored which factors promote group success in an offline and online logical task with adolescents. Five hundred and fifty high school students participated in the experiment and faced Raven's Advanced Progressive Matrices, first by themselves using the computer, then in a group. Groups interactions could have been computer-mediated or face-to-face, and the participants were randomly assigned to one of the two experimental conditions. Results suggest that groups perform better than singles, regardless of the experimental condition. Among adolescents, online groups performance was negatively affected by participants' average perception of group cohesion, the difficulty of the problem, and the number of communicative exchanges that occur in the interaction. On the contrary, the factors that improve their performances were the average intelligence of the teammates, their levels of neuroticism, and the group heterogeneity in terms of social abilities. This work contributes to the literature with a comprehensive model of collective intelligence among young people.

**Keywords:** collective-intelligence; computer-mediated-communication; group performance; problem-solving

## 1. Introduction

Collective Intelligence (CI) is the ability of groups to show higher levels of intelligence than their members [1]. It is the outcome of group problem-solving reasoning, that is usually more effective than the reasoning of one individual. Indeed, groups can efficiently and effectively process a large quantity of information, more than singles [2]. In the past few years, collective intelligence and individual intelligence received much attention from the scientific literature. If individual intelligence was defined as the ability of human beings to solve a wide variety of tasks [3], collective intelligence was considered as a general factor capable of explaining the "group's performance on a wide variety of tasks" [4]. Currently, the scientific evidence about the CI general factor are many, and they are supported by a meta-analysis carried out on 13 empirical studies [5]. The present research aims at analysing CI phenomenon, and specifically the outcomes of the group problem-solving reasoning, within both face-to-face and online groups composed of students who are facing logical–mathematical tasks.

According to the most up-to-date lines of study, CI is an emergent property of groups that results from both bottom-up and top-down processes [6]. The bottom-up processes involve the member characteristics that contribute to enhancing collaboration in group problem-solving reasoning. The top-down processes, instead, include the group structure and the norms that regulate collective behavior to improve the quality of members' coordination and group problem-solving reasoning outcomes. In particular, the most recent model of CI shows how three different factors explain about 43% of the group performance variance. The first is a top-down factor, namely the variance of the conversational turnover. The second and the third are two bottom-factors: the proportion of women in the group and the average of members' abilities in the theory of mind [4]. Other studies indicate that the average of group members' intelligence is a fundamental bottom-up factor in explaining the CI variance [7].

Although there is a lot of empirical evidence about the CI general factor, some scholars have tried to resize its effects on groups' problem-solving performance [7,8]. In particular, a recent re-analysis of the four main empirical studies in the field of CI [4,9–11] does not support the hypothesis of a general factor able to explain the performance variation across a wide variety of group-based tasks [8]. Studies carried out in the online environment show that it manifests differently depending on context [9]. Furthermore, from the literature, it can be seen that different factorial models could be suitable to explain the group problem-solving performance across different kinds of problems [8,12]. In this regard, Lam (1997) [13] argues that the structure of the task affects both the quality of group communications and decisions. In his studies, the author takes into account conjunctive, disjunctive, and additional tasks. Steiner (1972) [14] identified and described these three types of task structures. In the additive task, group performance is determined by the aggregation of individual effort [13]. Each group member has the same responsibilities and information, and they have to maximize his or her own personal performance to increase the overall group achievement [15]. In a disjunctive task, the group selects one optimal solution from an array of solutions proposed by individual group members [14,16]. Success in this kind of task is influenced by the performance of the members who make the most significant contribution. In a conjunctive task, no group member has enough information to solve the problem individually. Therefore, the successful decision can only be achieved when all the group members maximize their efforts [13]. In this kind of task, a group solves a problem only when all of the information held by individuals are merged in a single Collective Mental Map. Laughlin (1980) [17] proposed another noteworthy way to categorize the group-based tasks according to their structure. In particular, he suggested that group tasks might be placed on a continuum between intellectual and judgmental. Intellectual tasks are problems with only one correct solution, and that are demonstrable and testable. Conversely, judgmental tasks are problems with more than one acceptable solution (e.g., aesthetic judgment or juries deciding on guilt or innocence). Laughlin and Ellis (1986) [18] identified four conditions characterizing the intellectual group-based tasks. The first condition concerns the group members agreement about the proposed solution. Intellectual tasks usually favor teammates' agreement. The second condition regard the information available to the groups' members that must be sufficient to solve the problem. According to the third condition, everyone in the group can recognize the correct answer, so enough information must be provided to the members who do not know it. Finally, the fourth condition assumes that for the correct resolution of an intellectual group based-problem, the members who know the correct solution have enough abilities, motivation, and time to demonstrate it to the teammates.

In short, there are a lot of factors involved in the explanation of the variance of teams' performance in group problem-solving reasoning. In addition to top-down and bottom-up factors [6,7], the context in which the group work, the cognitive processes underlying the social problem-solving reasoning [1], and the structure of the task that has to be solved [8,13], appear as drivers of groups' performance. Furthermore, studies in the field of CI show how the difficulty of the task that the group is solving affect their performance. In particular, at the increase of the groups-based problems difficulty, the groups'

members' cooperation gets higher [19]. Thus, it is possible to hypothesize that different factorial models are needed to explain the variance of the CI across the wide variety of group-based tasks.

*The Group Problem-Solving*

The problem-solving abilities are generally considered as an appropriate indicator for the evaluation of the individuals' intelligence quotient [3]. A problem can be described as a situation where there is a gap between an initial state and a desirable condition; solving a problem means finding a way to fill that gap. Thus, problem-solving is a behavioral and cognitive process thanks to which people, developing many solutions for a specific problem, increasing the probability of selecting the most effective one [20].

According to Hayes (2013) [21], the process of problem resolution consists of two phases: understanding the nature of the problem and finding the correct strategy to solve it. During the first phase, a person develops an internal representation of the problem, identifying the goal, the initial state, the available tools, and the possible obstacles [21]. The internal representation of the problem is the medium by which reasoning takes place. It is subjective, so it is not precisely the same reproduction of the actual problem [21]. Mental representations can be represented, through drawn or written schemes. These kinds of representations are called "external representations" [21]. The second phase regards the cognitive and behavioral process that the problem-solver applies to reach the desired state. During this step, he examines the space of the problem, and implements strategies to reach the goal [21]. Hayes (2013) [21] identified four major strategies suitable to resolve a problem: (1) trial and error, (2) proximity methods, (3) fractionation methods, and (4) knowledge-based methods. The trial and error strategy involves the recursive evaluation of the whole process that led to the problem solution, exploring all the ways identified to solve it. The proximity strategy involves a systematic approach for the task resolution: the problem-solver getting closer to the goal by progressing step by step. The splitting strategy provides the subdivision of the main objective into sub-goals; according to this strategy, the problem-solver reaches the main scope solving the sub-goals one by one. Finally, according to the knowledge-based approaches, the problem-solver exploits knowledge stored in his memory to choose the best resolution strategy. The four resolution strategies are not mutually exclusive; thus, during the problem-resolution process, a person can first implement a strategy, then change it for another.

Once the strategy is executed, and its outcomes are evaluated, the problem-solver may start the consolidation process, during which the problem-solver is engaged in reflecting on the method used to solve the problem. The consolidation process plays a fundamental role in the learning process activated during problem-solving reasoning. Indeed, it allows for the creation of schemes that could drive the problem-solver in the representation of future problems [21].

People can implement problem-solving in a group or alone. When this process takes place within a group, members have to merge their efforts to find the correct solution [20]. Four factors can affect the outcomes of the group problem-solving: (1) group task, (2) group structure, namely the internal team organization, (3) group processes, that is the persuasive interactions occurring among members, and (4) group product [22].

Many studies have been carried out to compare group and individual problem-solving performance, but they reached different results. Indeed, while some scholars have concluded that groups negatively influence individuals' performance [14,23–25], others argue that groups, usually, outperform individuals [4,10,26]. Among those scholars who concluded that the group hinders the individual performance, Steiner (1972) [14] theorized that if groups produce less than expected, they are composed of members unable to find the correct solution or who fail to recognize the correct answer given by a teammate. Latané, Williams, and Harkins (1979) [25] shed light on the effect that groups have on individual effort. Comparing the individual and group performance, they found that when the participants are in the group condition, they reduce the effort invested in the clapping activity from a minimum of 28% to a maximum of 68%. This loss of individual effort in a group situation was called

"social loafing." Therefore, these results support the classical finding who showed how the effort in teams is reduced with the increasing of the size of the group [23,27].

At the same time, a lot of research does not support the idea that groups hinder the performance of single individuals. In an old study, Shaw (1932) [28] compared the performance of university students in solving logical problems (i.e., mathematical puzzles) working alone and in teams. The author suggests that, in mathematical puzzles, groups perform better than individuals. Lorge and Solomon [29], starting from the evidence of Shaw (1932) [28], proposed a mathematical model in which they identified, as a principal driver of the group performance, the ability of their members in recognizing the correct answer proposed by teammates. Furthermore, studies with both adults and college students showed that groups are more efficient and capable of providing better quality solutions than single individuals since they join together their knowledge and abilities [30–32]. In particular, in an experiment that involved university students, Laughlin and Bonner (1999) [33] found that groups can solve tasks effectively when problem-solving reasoning requires the elaboration of a large amount of information. Straus and Olivera (2000) [34], in their review about the effectiveness of online working-groups, have pointed out that group problem-solving can be a powerful learning tool to improve students' skills and knowledge. The advantage of group problem-solving in learning was also verified in children who were facing mathematical and logical problems [35–37].

Recent and not so recent research on CI found that the group can boost the performance of the individuals and that the collective interaction increases the knowledge of the group members [4,26,38].

Understanding how to exploit the full potential of the CI in the educational field and the online environment would be of great interest, in the light of the recent spread and use of online educational platforms. Indeed, the processes underlying CI, that determine its effectiveness in the face-to-face groups, are retained in online environments, where the interactions are computer-mediated [10,39]. However, much of the research in the field of CI focused mainly on adults, failing to investigate groups composed of young peers. Thus, this work aimed at investigating the CI phenomenon, and group problem-solving reasoning outcomes, in teams of adolescents who perform logical–mathematical tasks. First of all, it will be controlled if the groups hinder or boost the individuals' performance. Secondly, through the introduction in the experiment of two experimental group conditions, face-to-face and computer-mediated-communication, the purposes of this research was to evaluate the impact of the online environment on groups' performance. Thirdly, the influence of the intelligence of the group members on group problem-solving reasoning was analyzed. According to results proposed by Bates and Gupta (2017) [7], the teammates IQ should be positively correlated with the group-problem solving reasoning outcomes (to group members' high average IQ corresponds more effective solutions). Fourthly, since the sample was composed of schoolmates, the effect of the participants' sense of community on CI was evaluated. The generalized sense of community is defined as the group members feeling to belong. In particular, the sense of community can be considered as the group members awareness that teammates care about them. Individuals with a high sense of community are sure they are essential for the group, and they share the belief according to which they believe that their needs will be met thanks to their commitment to the group [40]. The empirical research in the field of CI usually involved samples of participants who did not know each other; thus, the effect of factors such as groups' cohesion or participants' sense of community on CI were poorly controlled. Therefore, here we hypothesized that a student's high sense of community may hinder the group performance. Indeed, according to literature, groups of friends have two main goals to achieve: on the one hand, they have to solve the problems assigned; on the other hand, they have to work to maintain the positive relationship between teammates [41]. For this reason, it was hypothesized that teams characterized by a higher sense of community will perform worse than groups not particularly cohesive. In light of this hypothesis, it was also assumed that groups characterized by greater conversational flow will achieve worse results in the proposed logical–mathematical group tasks. Indeed, it is possible to suppose that when the conversational turnover is intense, the groups are interacting to maintain the relationship among members, and not only to solve the logical–mathematical problems they are facing. Finally,

a model to isolate the main factors able to explain the variance of the online groups' problem-solving performance is presented.

## 2. Hypothesis

**Hypothesis 1 (H1).** *Groups outperform individuals in mathematical–logical problems. According to most of the literature about CI, groups can harness individuals' performance, showing a greater ability in the resolution of a wide variety of problems [1,4,10,11].*

**Hypothesis 2 (H2).** *Performances of the groups in computer-mediated (CMC) experimental condition do not significantly differ from the ones of the participants involved in face-to-face (FtF) interactions. According to some evidence about CI, the phenomenon should not be affected by the environment in which the group problem-solving process takes place [39].*

**Hypothesis 3 (H3).** *Groups' members' intelligence (IQ) is positively related to CI. According to empirical evidence [7], individuals' IQ is a determining factor that affects teams' performance in the resolution of group-based tasks. Therefore, it is hypothesized that in a learning environment, groups members' IQ positively affect the outcomes of the group problem-solving reasoning that teams implement to solve logical—mathematical problems.*

**Hypothesis 4 (H4).** *A high sense of community felt by groups members impairs teams performance in solving mathematical and logical problems. The empirical experiments carried out in the study of CI usually involved samples of adults who unknown each other. The CI studies implemented in educational environments never controlled the effects of group cohesion on group problem-solving outcomes. Here, it is hypothesized that group cohesion may negatively affect CI. Indeed, according to literature, a group of friends are steadily engaged in maintaining positive relationships among teammates [41]. Therefore, it is possible to guess that the commitment necessary to achieve the affiliation goals decreases the involvement in the intellectual objectives (the resolution of the logical–mathematical task assigned by the experimenters).*

**Hypothesis 5 (H5).** *A high communication flow among teammates decreases the effectiveness of groups' problem-solving reasoning. According to recent empirical evidence, the variance of the group's conversational turnover positively affects CI [4,10,11]. The present experiment does not purpose an analysis of groups' conversational turnover variance. Still, it is aimed at verifying if a high flow of information among teammates can negatively influence CI manifestation. It is hypothesized that when the number of groups' interaction is higher, members are engaging not only in mathematical–logical problem resolution but also in maintaining positive relationship among teammates. For this reason, it is assumed that a high flow of information among groups members hinders social problem-solving reasoning.*

**Hypothesis 6 (H6).** *The difficulty of the logical–mathematical problem which the groups are solving negatively affect their performance. Since previous research has shown how the difficulty of the problems increases groups cooperation [19], and since previous studies have put in light how the structure of the task influences group performance [13], the present study proposes the check of the effect that the difficulty of the task has on teams performance.*

## 3. Materials and Methods

### 3.1. Sample

The participants of the experiment were 563 high school students of Tuscany. Of the total sample, 460 were females and 103 were males. Students from each grade were included in the study. Thus, participants' age ranged between 14 and 19 years old. In particular, the average age of the students was 15.78 years (S.D. 1.50). The sample attended a Human Science High School, which offers an education in the humanities.

In the study, only those students who met the criteria for participation were included. The inclusion criteria were: fully understanding the Italian language; not satisfying the diagnostic criteria for developmental disorders; voluntary involvement; to have signed the informed consensus form (when participants were minors, the informed consensus was signed by legal tutors). Only 13 students (2.63% of the initial sample) did not fulfill the selected criteria. Thus, the final sample of the research was composed of 550 participants (Age $M = 15.62$ years $S.D. = 1.48$ years; 449 females, and 101 males).

The experiment was carried out following the guidelines of the Italian Psychological Association (AIP) in a manner that respected ethical and privacy issues.

### 3.1.1. The Psycho-Social Survey

At the beginning of each experimental session, a psycho-social questionnaire was administered to the participants, to control the possible effects of members' characteristics on group performance. The self-report survey was composed of two sections: a demographics section and a psychological one. First of all, data about gender and age of participants were collected, while the second section was devoted to assessing the following psychological characteristics.

- *Personality traits*: the I-TIPI inventory test ($\alpha = 0.59$) [42] is based on the Big Five factors Personality model purposed by Costa and McCrae (1992) [43]. According to this model, personality is composed of five different dimensions, namely extroversion, agreeableness, conscientiousness, neuroticism, and openness. Therefore, the I-TIPI test is composed of five sub-scales useful to assess the factors included in the Big Five Model. Therefore, the personality test adopted was composed of ten items through a seven-point Likert scale (1 = strongly disagree, 7 = strongly agree). Each of the five dimensions assessed through the I-TIPI inventory test was measured by means the combination of two items.

- *Group members sense of community (Group cohesion)*: the sense of community (SOC) was measured using the Classroom and School Community Inventory (CSCI) ($\alpha = 0.84$) [44], which assigns two separate scores to participants: one for the *Learning Community* ($\alpha = 0.87$) and one for the *Social community* ($\alpha = 0.90$). The scale was composed of 10 items, 5 for each sub-scale, on a five-point scale (1 = strongly agree, 5 = strongly disagree). Therefore, high scores on this scale denote a low level of perceived SOC, while low scores indicate a high level of it. The literature defined the generalized sense of community as the group members feeling of belonging to a group [40]. Therefore, the sense of community can be considered as a proxy for the study of groups' cohesion.

- *Social abilities*: the Italian version of the Reading the Mind in the Eyes test (RME) ($\alpha = 0.605$) [45] has been administered to measure participants' social abilities. RME is a widely used test for the assessment of the theory of mind, namely the ability of a person to understand what another individual thinks and feels. Here, we adopted this test for the evaluation of students' social abilities because it was included in the pioneering empirical study on CI [4] and in its more recent replication [7]. The RME test was composed of 36 images displaying the eyes of different people that present a variety of emotions, and the participants were asked to select which emotion was shown, with a choice of four different options.

### 3.1.2. Stimuli

The Raven's Advanced Progressive Matrices (Set II) was chosen as the research stimuli in this research for three main reasons. First of all, the same test, and the same partition of matrices for individual and group conditions, were used by Woolley, Chabris, Pentland, Hashmi, and Malone [4] in their pioneering work on the CI. Moreover, it became part of the *collective intelligence test battery online tool* [10], a validated research procedure for the study of the CI. Secondly, RAPM is one of the most widely-used tests for the assessment of the intelligence quotient, and it is not affected by participants'

cultural backgrounds [46–48]. Finally, Raven's Progressive Matrices maintains its validity in the digital form [49].

The experiment outlined here consisted of two phases. During the first phase, the students individually faced the Raven's Advanced Progressive Matrices, while during the second phase, they were divided into teams to solve similar problems within a group. During both phases, the experiment was carried out using a digital version of the Raven's Advanced Progressive Matrices (RAPM). The software for the digital test was specifically developed for this study. Only the matrices included in the original paper test were used in the RAPM digital version. The full RAPM test consisted of 36 matrix puzzles. To solve each puzzle, participants had to identify between eight different options, the missing element of a grid that completed the pattern. In the present research, the first phase provided the assessment of each participant's intelligence through the 18 odd-numbered Raven's matrices. Instead, for the assessment of the collective intelligence, groups solved the remaining 18, even-numbered matrices of the RAPM test.

### 3.1.3. Procedures

The experiment lasted two weeks, and was conducted in the school during the lesson time and was performed in two steps. In the first step, participants were asked to fill a self-report survey. During the second step that occurred one week after the first, participants completed two trials: an intelligence assessment task, carried out individually, followed by a group task. In both tasks, a time constraint was included, giving participants 15 min for each trial (i.e., 15 min for both individual and group steps). In Figure 1, the user interface shown to participants during the individual tasks resolution phase can be seen.

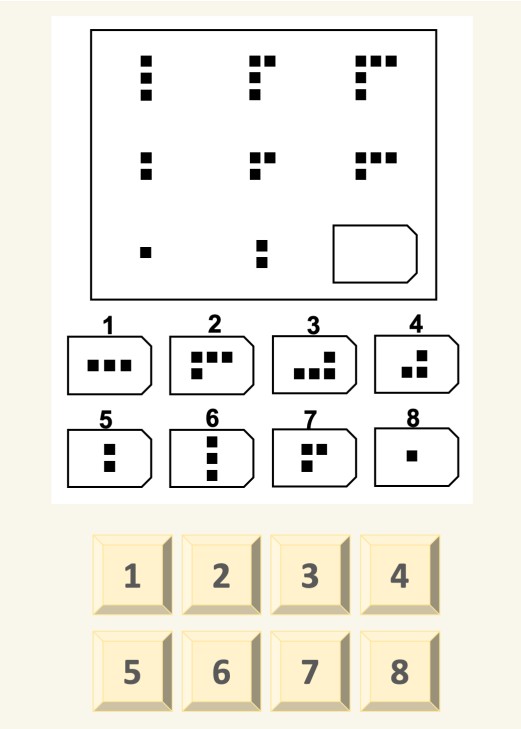

**Figure 1.** Users' interface for individual task. Users' interface of the Raven's Advanced Progressive Matrices test (RAPM) that participants faced individually. The buttons at the bottom of the interface correspond to the eight answer options for the matrix resolution.

Regarding the group experimental phase, two conditions were implemented: computer-mediated-communication (CMC) and face-to-face (FtF). Before the beginning of the experiment, participants from each class were randomly divided into groups of five members, and each group was randomly

assigned to one of the two experimental group conditions. At the end of the experiment 57 groups completed the task in the FtF condition (230 females and 55 males) and 53 groups completed the task in the CMC condition (219 females and 46 males).

In the CMC condition, five participants gathered in a group were seated at PC stations equipped with a tablet and a pair of earphones. Participants were physically isolated from each other. Using the tablet, participants could see the matrices, evaluate the possible answers, and select the chosen solution. Through the earphones, they could interact with their teammates. Figure 2 shows the users interface for the computer-mediated-communication group condition. During the whole group experimental phase, participants saw the same users' interface, and the same Raven's matrix, of their teammates. It can be noted, the CMC condition interface Figure 2 is not equipped with a chat, so, participants within this condition interacted only vocally through headphones and microphones. Google Hangouts was used to allow only voice communication between participants, indeed, no webcam was present on the PC stations. The group could go ahead with the following matrix only if at least three out of five members picked the same answer. Otherwise, the system kept showing the same matrix until the majority agreement criteria was met.

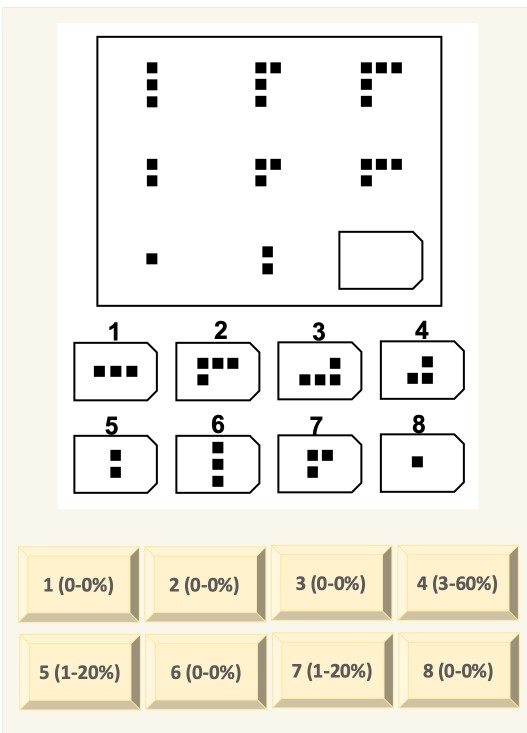

**Figure 2.** Users' interface for the task in the computer-mediated-communication (CMC) condition. Users' interface of the Raven's Advanced Progressive Matrices test (RAPM) that participants have faced in the computer-mediated-communication experimental condition (CMC). In the buttons at the bottom of the interface, other to be represented the number corresponding to the eight answer options for the matrix, there are both the number and the percentage of teammates who have given that answer.

In the FtF condition, a group of five participants was placed around an interactive whiteboard where each matrix was projected. Each member of the group could speak with the others to find the correct answer and reach a majority agreement. Once the approval was obtained (i.e., $\frac{3}{5}$ of the team agreed), the group should communicate its choice to the researcher, who annotated it trough a special panel in the software, together with the percentage of agreement in the group. Figure 3 reports the software interface for the face-to-face group condition.

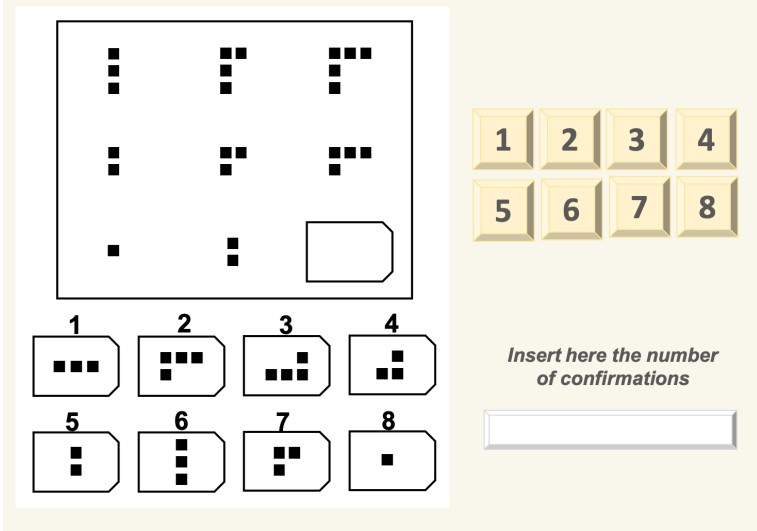

**Figure 3.** Users' interface for the task in face-to-face (FTF) condition. The user interface of the panel used by researchers to save the data during experimental sessions of face-to-face groups (translated in English from Italian). Buttons 1–8 correspond to the eight answers options for the matrix, and the space under the buttons was used to take note of the number of members who agreed with the given solution.

*3.2. Analysis*

After the data collection, the analysis was performed. First of all, the descriptive analyses were carried out to verify the preconditions necessary for subsequent inferential analyses. In particular, we calculated the mean, standard deviation, skewness, and kurtosis of each measured factor. The skewness and kurtosis indexes were the indicators for the normality of factors. We also controlled for the normality of residuals.

After descriptive analysis were performed, inferential analysis was conducted. The Student *t*-test analysis was used for evaluating individuals' and groups' performance (H1). In these analyses, we compared the individual and group performance by considering only the scores obtained by the same participants before in the individual task and after in the group task. For this reason, it was possible to perform two paired-sample Student *t*-tests, one confronting individuals' and CMC groups' outcomes, and one comparing individuals' and FtF groups' performance. For what concerns the second hypothesis of the current study, an unpaired Student *t*-test was used to compare face-to-face and computer-mediated-communication experimental conditions (H2). The ratio between the correct answers and the number of matrices faced during the 15 minutes given for each prove was used as a dependent parameter for both Student *t*-test analyses. This decision was made based on Szuba's (2001) [50] theory about group performance, according to which CI must be parameterized as a probability function to solve problems in a specific timescale.

Finally, in order to verify the third, fourth, and fifth hypotheses, we utilized a *Generalized linear Mixed Model* (GLMM), including among its factors both the teammates' average RAPM score, participants' average sense of community score, and conversational turnover. Furthermore, in order to control the effect of other important factors for the explanation of the variance of groups' CI, we included teammates' variability in rme score; teammates' average neuroticism score; and the difficulty of the problem in the model. The variable difficulty of the problem was computed taking advantage of the RAPM design to verify its impact on groups' problem-solving outcomes. Indeed, the RAPM test, in each step provided the presentation of an even more complex problem. In other words, each matrix was easier than the next, so the first matrix was significantly easier than the latter. By dividing the eighteen matrices of which both RAMP tests were composed (individual and group RAPM test) into three categories, this work provided the analysis of CI on three different levels of mathematical–logical problems' difficulty: easy, medium–hard, and hard. The GLMM was performed

only for the CMC condition, because of the experimental framework has foreseen the measurement of the conversational turnover only in CMC groups. Since the literature has shown how conversational turnover is one of the main factors influencing group performance, we chose to not propose a GLMM for the FtF condition, as it did not take this factor into account. However, since no differences were identified between the CMC and FtF experimental groups, it is possible to hypothesize that the CMC model also fit the FtF condition.

### 3.3. Results

Figure 4 shows the comparison between the participants' performance in the individual RAPM test and the FtF groups' ability in the resolution of the same task. We compared the individual and group performances including in the sample only the score obtained by the same participants in the group and in the individual performance. As can be seen in the box-plot in Figure 4, the two distributions differ widely from each other.

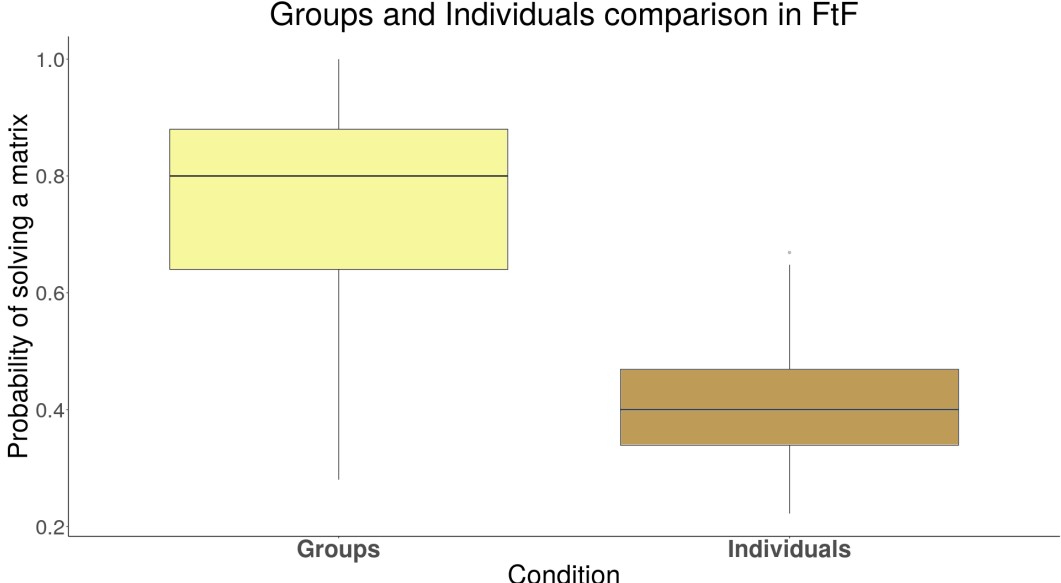

**Figure 4.** Comparison between individuals' and FtF groups' performance. A box-plot is shown, comparing the performance in Raven's Advanced Progressive Matrices (RAPM) of the groups involved in face-to-face condition (FtF) and the score obtained in the test by teammates when they worked individually (teammates' average RAPM score). The horizontal lines in the box indicates the median values, respectively $Med = 0.8$ for groups and $Med = 0.4$ for individuals.

The *t*-test analysis performed found significant difference between the performance achieved by individuals and FtF groups ($t_{(56)} = 14.674, p < 0.001, d = 1.91$). In particular, the mean of the individual RAPM score obtained by participants in the FtF condition ($M = 0.41, S.D. = 0.11$) was significantly lower then the one obtained by groups in the same task ($M = 0.74, S.D. = 0.18$). Therefore, according to first hypothesis, in FtF condition groups outperform individuals in problem-solving outcomes. Similar results were achieved by participants involved in CMC experimental condition. As can be observed in Figure 5, groups show higher score than their members in online environments. The paired sample Student *t*-test analysis highlights a significant difference between individual and CMC groups' performance ($t_{(109)} = 1.39, p = 0.166, d = 0.266$). Indeed, teammates' average RAPM score ($M = 0.392, S.D. = 0.096$) results were significantly lower than the results obtained by groups ($M = 0.696, S.D. = 0.19$) in the same test. Both results obtained from the paired sample Student *t*-tests that were performed to compare individuals and group problem-solving performance support the first hypothesis of the current study (H1).

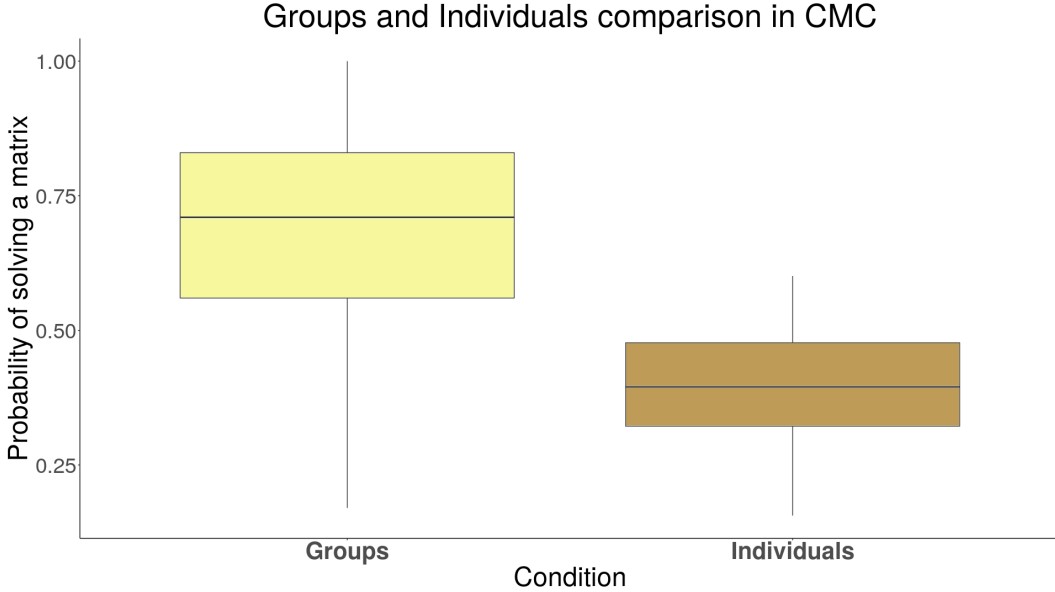

**Figure 5.** Comparison between individuals' and CMC groups' performance. A box-plot is shown, comparing the performance in RAPM of the groups involved in computer-mediated-communication (CMC) and the score obtained in the test by teammates when they worked individually (teammates' average RAPM score). The horizontal lines in the box indicates the median values, respectively $Med = 0.733$ for groups and $Med = 0.386$ for individuals.

Hypothesis 2 predicts no difference in the performance of CMC and FtF groups. A third Student *t*-test was performed to verify the second hypothesis. The results are presented in Figure 6. The box-plot shows that the two distributions barely differ from each other. The *t*-test analysis found no significant difference between the performance achieved by groups that completed the task in the CMC condition and the ones who worked in FtF conditions ($t_{(109)} = 1.39, p = 0.166, d = 0.266$). Therefore, the *t*-test results support H2.

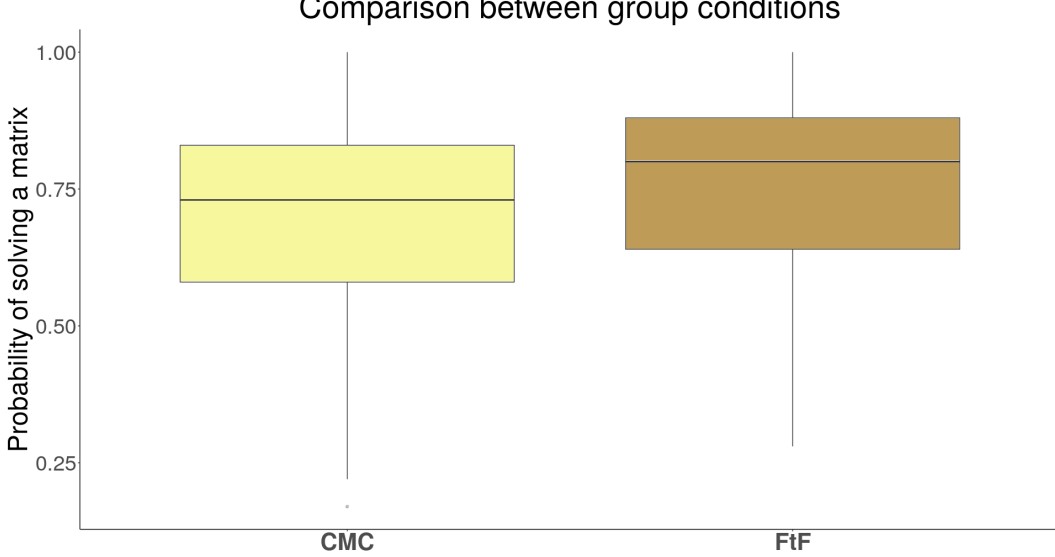

**Figure 6.** Comparison between performance of groups in CMC and FtF conditions. A box-plot is shown, comparing the performance in RAPM of the groups involved in computer-mediated-communication (CMC) and groups involved in face-to-face condition (FtF). The horizontal lines in the box indicates the median values, respectively $Med = 0.733$ for CMC and $Med = 0.8$ for FtF.

A *Generalized Linear Mixed Model* was used to control the effect of some factors on CI in groups within the CMC condition, and to verify the third, fourth, and fifth hypothesis of the current study. In particular, the teammates' average RAPM score, participants' average sense of community score, conversational turnover, teammates' variability in RME score (i.e., RME standard deviation), teammates' average neuroticism score, and difficulty of problem are the six factors included in this inferential analyses. The results of the GLMM model are presented in Table 1.

**Table 1.** Generalized Linear Mixed Model. Effect of groups, members, and tasks characteristics on collective intelligence.

| | **Akaike** | **F** | **Df-1(2)** | **Model Precision** |
|---|---|---|---|---|
| Best Model | 17,453.795 | 100.412 *** | 7(3662) | 76.4% |
| **Fixed Effects** | | | | |
| **Factors** | **F** | **Df-1(2)** | **Coefficient ($\beta$)** | **Student t** |
| Teammates' Variability in RME Score | 26.761 | 1(3662) | 0.206 | 5.173 *** |
| Conversational Turnover | 8.639 | 1(3662) | −0.003 | −2.939 ** |
| Teammates' Average Neuroticism Score | 19.356 | 1(3662) | 0.175 | 4.400 *** |
| Teammates' Average CSCI Score (SOC) ‡ | 19.656 | 1(3662) | 0.091 | 4.434 *** |
| Teammates' Average RAPM Score | 103.351 | 1(3662) | 4.942 | 10.166 *** |
| Difficulty of the Task † | 262.929 | 2(3662) † | −2.838 †† | −22.697 *** †† |
| | | 2(3662) ††† | −1.220 ††† | −13.252 *** ††† |

*** = $p$ <0.001, ** = $p$ <0.05, † = easy as reference value, †† = medium-hard, ††† = hard, ‡ = high scores on this scale indicate low sense of community.

As can be seen from Table 1, groups' performance in solving logical–mathematical problems appears to be influenced by both individuals' features, groups' characteristics, and task difficulty. For what concerns the individual features, H3 assumed that teammates' average RAPM score was positively related with the CI. The results confirm the hypothesis. Furthermore, the probability to choose the correct answer to the matrices proposed in RAPM test is higher in groups with more significant levels of teammates' average neuroticism. Regardless of the groups' characteristics measured in the current study, H4 predicted worst performances in groups with higher levels of teammates' average sense of community (SOC). GLMM results support the H4 hypothesis. Indeed, since higher scores obtained in the Classroom and School Community Inventory test (the questionnaire used here for the evaluation of SOC) indicate lower levels of perceived SOC, the positive relation reported in the Table 1 between such factor and CI means that lower levels of perceived sense of community increase groups' problem-solving performance. In line with H4, H5 assumed that high communication flow among teammates decreases the effectiveness of groups' problem-solving reasoning. As can be observed in Table 1, GLMM also confirms this hypothesis. Furthermore, since the literature proposes very consistent results regarding the relation among groups' members' social abilities and CI, it was included in GLMM to control its effect on groups' RAPM Score. Results indicate an interesting and positive relation between groups' problem-solving performance and groups' members' social abilities heterogeneity. Finally, since according to literature the structure of the task is a driver of group performance, it have been included in the GLMM a three levels factor representing the difficulty of the problem that the groups solved. As can be observed from the Table 1, at the increase of the matrix's difficulty, the groups' performance get worse. Thus, the GLMM corroborate the sixth hypothesis.

## 4. Discussion

This study proposed an analysis of the collective intelligence (CI) phenomenon both in real and online environments. In particular, empirical evidence on problem-solving implemented by groups of adolescents who were facing logical–mathematical tasks was presented. In-sum, it was found that groups have the potential to boost and enhance individual problem-solving performance, both in

real and online environments. The results of this study are in line with the empirical research in the field of collective intelligence, namely the ability of the group to show a higher intelligence than its members [1].

In the experiment presented here, participants were randomly assigned to two different experimental conditions: face-to-face groups (FtF) or computer-mediated-communication groups (CMC). All the participants faced the Raven's Advanced Progressive Matrices two times: the odd-numbered ones individually and even-numbered ones cooperating with four other classmates. Furthermore, adolescents completed a psycho-social survey that was administered to collect crucial information about their individual characteristics (i.e., gender, age, intelligence quotient (IQ), sense of community, personality traits, and social abilities). The research has already shown how groups' members IQ and social abilities are drivers of group performance. The results partially confirm the literature. Indeed, on one hand, it was found that at the increase of the teammates' average IQ, the groups' performance gets better; on the other hand, no significant effects were found in the relation between social abilities and CI. In this study, we detected a significant effect of the heterogeneity in members' social abilities in boosting groups' problem-solving performance. However, the empirical research in the field of CI usually involved samples of adults. Thus, the idiosyncrasy detected between the literature and the present results about teammates' social abilities may be ascribed to the age of the experimental participants of this study. Indeed, the research in the field of psychological development has repeatedly proven how the abilities in the theory of mind are not yet completely developed during the earlier years of adolescence. It may be possible that the underdevelopment in the theory of mind abilities have modified the influence of members' social skills on group problem-solving performance. For what concerns groups' members' sense of community, since the research in the field of CI usually involves participants who are randomly recruited and unknown each other, no empirical study has controlled the effect of this factor. According to the literature on the group's performance, it was hypothesized that a higher sense of community perceived among teammates decreases the effectiveness of group's problem-solving reasoning. Indeed, when a group of friends were solving a problem, members' attention and motivation were employed both in seeking task solution and maintaining positive relations with teammates [41]. Therefore, the commitment placed by the groups composed of friends in maintaining a positive relationship among the members may hinder their engagement in solving other problems (e.g., the logical–mathematical ones). The results corroborate this hypothesis. Thus, during the group task, participants with a high sense of community may have engaged some their cognitive resources to reach affiliation goals (i.e., maintain positive relationships with classmates), leaving aside the solution of the logical–mathematical problems. This, as a process, may be able to explain the negative relation between members' sense of community and CI that was found. Another finding that may corroborate the interpretation of this result is that higher the levels of groups' conversational turnover were, the worse their performance was. Therefore, it is possible to suppose that experimental groups which were composed of classmates have lingered on interactions not only directed to the matrices resolution but also finalized to something else, perhaps the pursuit of affiliations goals. Concluding, since participants in the experiment was classmate, their need of affiliation and sense of community may have hindered their engagement in solving the mathematical problem.

We administered a survey for the detection of personality traits. By checking the effect of the teammates' average personality traits on groups' problem-solving performance, it was found that at the increasing of teammates' average neuroticism, the groups' performance was better. Finally, the effect of the difficulty of the problem that the groups had to solve on their performance was controlled. Intuitively, the complexity of the logical–mathematical task negatively affect the groups' problem-solving outcomes.

In sum, the results of this study, besides showing how a group can boost high school student performance in both logical and mathematical tasks, propose a model of collective intelligence suitable to explain adolescent group performance in online group problem-solving. The factors included in

such model are: teammates' average social abilities, teammates' average IQ, teammates' average neuroticism, teammates' average sense of community, groups' conversational turnover, and difficulty of the problem that the teams are solving.

Some limitations could be relevant to this work. First of all, it has not been possible to gather data about speaking variance in the FtF condition, namely the actual number of speaking turns of each participant. This variable could have represented a precious source of information given the school peers context involved, moreover, it represents a parameter evaluated in the vast majority of experiments in CI [4,10,26]. Secondly, participants involved in the study represent a convenient sample and are heavily unbalanced in favor of the females' numbers. Future works may try to take into consideration these limits to improve the presented research.

Although the empirical-research about group performance modeling is still elusive, it is clear how the predisposition to form groups has been one of the factors that lead human beings to successfully compete in the struggle for survival during their evolution [51]. This attitude allowed humans to overcome complex problems, otherwise impossible for an individual [2].

The research described here provides some possible perspectives in the direction of exploiting CI, especially in the field of online education. The findings from this study suggest that CI principles could also be harnessed in online educational contexts. Indeed, the results presented indicate that small working groups could obtain better results than people working individually, both in the real and online environment. This could guide the design of the future implementation of e-learning platforms and school laboratories, even considering literature findings that link CI with increasing learning abilities.

**Author Contributions:** A.G. directed the project, E.I., F.S., and A.G. envisaged the problem, A.G., F.S., and E.I. designed the experiment, E.I. and A.G. developed the software used in the experiment, E.I., F.S., and A.G. conducted the experimental phase, A.G., F.S., and E.I. analyzed the data, F.S. and E.I. wrote the manuscript, all authors reviewed it. All authors have read and agreed to the published version of the manuscript.

**Funding:** This research received no external funding.

**Conflicts of Interest:** The authors declare no conflict of interest.

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
