# Peer review of "Adolescent’s Collective Intelligence: Empirical Evidence in Real and Online Classmates Groups"

_futureinternet, doi:10.3390/fi12050081_

Round 1

Reviewer 1 Report

Regarding the title, the "new model" expressed in the title is not clearly defined, the article is a case study, but a section is missing where the authors propose a model and it is validated based on the case study.

The introduction section at the beginning (up to page 3) is especially dense and complicated to follow the common thread, it would be better to structure it and include a comparative table or graph between models, which would facilitate its monitoring.

Section 1.1 is better structured, but perhaps part of this content or reference is not part of the "introduction" but of the "state of the art" or even of the "methodology", in section "3.1.3 Procedures". it should also be included in the "Discussion" section, corroborating or reaffirming the described literature.

Regarding methodology, a limitation may be the limitation of written language, only voice communication, where we do not know if they have the possibility of using shared desktops or interacting on the screen. The tools or the characteristics of the PC stations can be interesting to know for the CMC-FTF analysis, since for example, the current situation of non-classroom teaching in some countries is being compensated with a wide variety of tools that allow the CMC to be brought closer to the FTF.

With respect to the bibliography, it would be advisable to review the citations, since different bibliographic styles are used (Numbers and Author-Year).

Author Response

First of all, we want to thanks reviewers for their helpful and constructive advice. We tried to improve the manuscript according to their suggestions. In short, we have significantly modified each section of the paper, starting from the introduction and concluding with discussion. Furthermore, we tried to improve both images and captions. Finally, according to the suggestion provided by reviewer 3, we inserted a more in-depth discussion about the effect of the teammates’ sense of community on group performance. Following, we provide specific answers to each comment proposed by reviewers.

Our best wishes,

the authors.

Comments of the reviewer 1

Point 1: Regarding the title, the "new model" expressed in the title is not clearly defined, the article is a case study, but a section is missing where the authors propose a model and it is validated based on the case study.

Response: We have modified the title according to the reviewers’ observations.

Point 2: The introduction section at the beginning (up to page 3) is especially dense and complicated to follow the common thread, it would be better to structure it and include a comparative table or graph between models, which would facilitate its monitoring. Section 1.1 is better structured, but perhaps part of this content or reference is not part of the "introduction" but of the "state of the art" or even of the "methodology", in section "3.1.3 Procedures".

Response: We tried to propose a more structured introduction following the suggestions of all the reviewers, but we opted not to design tables to compare the mentioned theories. Reading and correcting this section according to all the recommendation, we have chosen to delete some literature results to proposes a more synthetic version of the introduction. In particular, the first part of the introduction was significantly shortened and simplified.

Point 3: It should also be included in the "Discussion" section, corroborating or reaffirming the described literature.

Response: We tried to improve the discussions providing the explanation of our results in the light of the literature in the field of CI

Point 4: Regarding methodology, a limitation may be the limitation of written language, only voice communication, where we do not know if they have the possibility of using shared desktops or interacting on the screen. à

Response: We add more details about users’ software interfaces and modality of communication in the “procedure” part of the paper

Point 5: The tools or the characteristics of the PC stations can be interesting to know for the CMC-FTF analysis since, for example, the current situation of non-classroom teaching in some countries is being compensated with a wide variety of tools that allow the CMC to be brought closer to the FTF.

Response: we improve the description of the tools used

Point 6: With respect to the bibliography, it would be advisable to review the citations, since different bibliographic styles are used (Numbers and Author-Year).

Response: We corrected both citations and bibliography.

Reviewer 2 Report

I suggest to improve the design of the figures.

I agree with the publication of the article after minor corrections.

Author Response

First of all, we want to thanks reviewers for their helpful and constructive advice. We tried to improve the manuscript according to their suggestions. In short, we have significantly modified each section of the paper, starting from the introduction and concluding with discussion. Furthermore, we tried to improve both images and captions. Finally, according to the suggestion provided by reviewer 3, we inserted a more in-depth discussion about the effect of the teammates’ sense of community on group performance. Following, we provide specific answers to each comment proposed by reviewers.

Our best wishes,

the authors.

Comments of the reviewer 2

Point 1: Improve figures 

Response: Improved.

Reviewer 3 Report

The article examines the concept of collective intelligence (CI) in a sample of adolescents engaged in a logical-mathematical task. It aims at comparing individual and collective results when groups either use computer-mediated or face-to-face interactions. It also aims to isolate the main factors that explain the ability of online groups to solve logical-mathematical problems.

Overall Comments

In general, the paper is well articulated. The literature review seems comprehensive and adequate. The text clearly presents the research hypotheses, and the methods used to validate them. The results and conclusion are interesting but their presentation could be enhanced. The weakness of the paper is found in the Discussion section. The section should be improved by developing on the implications of their findings and those obtained in similar researches. It must also address in more detail the impact that the choice of participants may have had on their results.

Major Comments

Lines

Comment

Section 2

Reformat text to have each hypothesis forming a block. Currently the explanation texts seem to be linked to the wrong hypothesis. Or maybe give the explanation first and then the hypothesis.

245

Remove “expressly developed for this study”, it looks as if you had “developed” some new figures for the study.

270-272

I understand the groups were composed of participants of the same class. Was it on purpose? What if they came from different classes? Does the proportion of participants knowing each other in a group could have changed the results? Discuss and provide references.

288-300

Rephrase and restructure the paragraph to make it clearer. Like “According to … we made this choice” or “In order to consider … we add such variable to the study.”

Figures

For figures 4-6, have you considered using box plots, instead of adapted histograms? It could provide very insightful information about your results (mean, median, quantiles, …).

315-326

Structure the text here in order to put the table for the discussion.

Discussion

The text must be revised/reformulated by a native English speaker.

Discussion

Consider restructuring it to highlight the relationships between results and the different factors/variables.

Discussion

The fact that the participants knew each other before the experience should have been highlighted from the start, and should be discussed in much more detail since this affected the results on more than one level.

Figures

Move Figures 5 and 6 to the Result section

334-339

Remove. It has all been said in the method section.

Table 1

Move the table to the Result section

339-341

“The finding support … work with known people…” That is interesting but it is not stated clearly in your hypotheses. Furthermore it affected your results so it is imperative you made it clear in the introduction, the method and the discussion section.

346-351

“These findings … logical tasks” same comment as above

373-374

Discuss interaction with the second variable mentioned above (total conversational turnover). Results may seem contradictory.

Minor Comments

Lines

Comment

6-8

Rephrase, split the sentence.

26

“different variables” do you mean different factors?

33

“should to take in more account” replaced by “should be taken into account”

35

“Heylighen 1999” Please use the reference style suggested in Journal’s instruction to author

46

The second way of building a CMM is through a division of labour within the group. ?

54

“Then, with Heylighen…” Make it clearer that it is the author of the last paper you cite [5]

62

Replace “that it is possible to suppose” by “it is possible that”.

79

“Steiner 1972…”  Repetitive with previous sentence

90

Remove “also”

96

“ … as a proxy to evaluate…” a proxy criterion?

109

“external” maybe “external representation”?

114

Is “fractionation” the best word to use here according to the wording later used?

124-125

Rephrase

130-131

“People … problem-solving process, …” replaced by “In a group problem-solving, ...”

143

“How already synthetically exposed, these” replaced by “As already mentioned, these”

144

“hinder” may hinder? Provide reference here, even if you repeat it a few sentences later.

148

Reference duplication (Steiner)

154

Ringelmann: add the reference number

163

Replace “greater efficiency” by “more efficient”?

168

“work groups,,” (,,)!

178

Replace “… CI ... environments” by “… CI is retained in online environments”?

192

“computer-mediated (CMC) interaction with face-to-face (FTF) interaction”

185

“No significance difference” should rather be “No significant difference”

205-206

High school spans differ between countries, rather provide the age of participants (range)

218-219

“characteristic, a psychosocial questionnaire was administered to all the participants”

225

“ … big five (OCEAN) model [44].”

244-247

Merge in one paragraph

248-251

Merge in one paragraph

252-259

Make it the first paragraph of the section

261

“This two phases experiment was made at school…”

261-262

Remove “was composed of two phases that”.

265

Use Figure instead of Fig. in your text

265-269

Merge in one paragraph

274

Replace “5 participants for” by “participants of”.

278

“least”, not “list”

Figure 4

Give again the definition of both groups in caption, state again the acronyms, and use acronyms in the graph.

316

The use of a multivariate model should have clearly appeared in the method section

323

“wide” instead of “width”

333-334

Remove “proposed here".

Table 1

Could the table be formatted to have Factors described in one line only?

354

What do you mean by “importance of the group”? Explain

358-359

“average members’ intelligence” really? Restate by rather referring to their RAPM scores.

365-366

“First of all, … performance decrease” Isn’t it obvious?

383-393

Consider using these lines in the conclusion section.

384

“the predisposition to form groups” Detail your thought, cite the reference.

386

“for a single individual”?  for the average individual?

Author Response

First of all, we want to thanks reviewers for their helpful and constructive advice. We tried to improve the manuscript according to their suggestions. In short, we have significantly modified each section of the paper, starting from the introduction and concluding with discussion. Furthermore, we tried to improve both images and captions. Finally, according to the suggestion provided by reviewer 3, we inserted a more in-depth discussion about the effect of the teammates’ sense of community on group performance. Following, we provide specific answers to each comment proposed by reviewers.

Our best wishes,

the authors.

Comments of the reviewer 3

Point 1:Reformat text to have each hypothesis forming block. Currently, the explanation texts seem to be linked to the wrong hypothesis. Or maybe give the explanation first and then the hypothesis. 

Response: Done.

Point 2: Remove “expressly developed for this study”, it looks as if you had “developed” some new figures for the study.

Response: We have chosen to improve the sentence explaining that we have developed the online version of the test, not including new matrices.

Point 3: Rephrase and restructure the paragraph to make it clearer. Like “According to ... we made this choice” or “In order to consider ... we add such variable to the study.” 

Response: We rewrote all the section results tiring to explain our chooses in data analyses.

Point 4: For figures 4-6, have you considered using box plots, instead of adapted histograms? It could provide very insightful information about your results (mean, median, quantiles, ...).  

Response: We didn’t consider before to present our results through a box plot. We appreciated the suggestion, so we have replaced bar-graphs with box plots.

Point 5: Table: Move the table to the Result section. 

Response: Done.

Point 6: 315/326 Structure the text here in order to put the table for the discussion. à Response: Done.

Point 7: Discussion: The text must be revised/reformulated by a native English speaker. à Response: Done.

Point 8: Discussion: Consider restructuring it to highlight the relationships between results and the different factors/variables. 

Response: We significantly modified the discussion section, structuring it according to results

Point 9: Discussion: 1) The fact that the participants knew each other before the experience should have been highlighted from the start, and should be discussed in much more detail since this affected the results on more than one level. à 2) I understand the groups were composed of participants of the same class. Was it on purpose? What if they came from different classes? Does the proportion of participants know each other in a group could have changed the results? Discuss and provide references. à 3) 339-341 “The finding support ... work with known people...” That is interesting but it is not stated clearly in your hypotheses. Furthermore, it affected your results so it is imperative you made it clear in the introduction, the method and the discussion section.

Response: We added the discussion of the effect of teammates’ sense of community on CI in all section of the paper. Specifically:

1) Introduction: Line 167- 178;

2) Hypothesis: Lines 194-211 (H4; H5);

3) Analysis: Line 335;

4) Results: Lines 187-195;

5) Discussion: Lines 430-450.

Point 10: Figure: Move Figures 5 and 6 to the Result section  

Response: Done.

Point 11: 334-339 à Remove. It has all been said in the method section.

Response:à Removed.

Point 12: 373-374 Discuss interaction with the second variable mentioned above (total conversational turnover). Results may seem contradictory.

Response: Discussed in lines 430-450

Minor Comments reviewer 3

Point 13: Line 6-8 Rephrase, split the sentence.  

Response: Done.

Point 14: 26 “different variables” do you mean different factors?

Response: We have corrected.

Point 15: 33 “should take in more account” replaced by “should be taken into account” à Response: Modifying the introduction according to the Reviewer 1 observations, we have removed all the sentence.

Point 16: 35 “Heylighen 1999” Please use the reference style suggested in Journal’s instruction to the author.

Response: Modifying the introduction according to the Reviewer 1 observations, we have removed this reference.

Point 17: 46 The second way of building a CMM is through a division of labour within the group. ?

Response: Modifying the introduction according to the Reviewer 1 observations, we have removed all this part of the paper.

Point 18: 54 “Then, with Heylighen...” Make it clearer that it is the author of the last paper you cite [5].

Response: Also, in this case, modifying the introduction according to the Reviewer 1 observations, we have removed this sentence from the paper

Point 19: 62 Replace “that it is possible to suppose” by “it is possible that”.  

Response: We have rewritten the sentence.

Point 20: 79 “Steiner 1972...” Repetitive with previous sentence à

Response: Removed.

Point 21: 90 Remove “also”.  

Response: Done.

Point 22: 96 “ ... as a proxy to evaluate...” a proxy criterion? 

Response: We have rewritten the sentence removing the incorrect word “proxy”.

Point 23: 109 “external” maybe “external representation”?

Response: Done.

Point 24: 114 Is “fractionation” the best word to use here according to the wording later used?

Response: Of course. We took the word from the original text of the author (Hayes, J.R. The Complete Problem Solver, 2 ed.; Routledge: New York, 2013. doi:10.4324/9780203062715., pp. 35).

Point 25: 124-125 Rephrase.

Response: Done.

Point 26: 130-131 “People ... problem-solving process, ...” replaced by “In a group problem-solving, ...”. 

Response: Replaced.

Point 27: 143 “How already synthetically exposed, these” replaced by “As already mentioned, these”. 

Response: Correcting the paper, we decided to delete all the sentence at issue

Point 28: 144 “hinder” may hinder? Provide reference here, even if you repeat it a few sentences later. 

Response: We did it.

Point 29: 148 Reference duplication (Steiner) 

Response: Corrected.

Point 30: 154 Ringelmann: add the reference number 

Response: Done.

Point 31: 163 Replace “greater efficiency” by “more efficient”?  

Response: Corrected.

Point 32: 168 “work groups,,” (,,)! 

Response: Corrected.

Point 33: 178 Replace “... CI ... environments” by “... CI is retained in online environments”?

Response: We rewrote all sentence

Point 34: 192 “computer-mediated (CMC) interaction with face-to-face (FTF) interaction” + 185“No significance difference” should rather be “No significant difference” à Response: We have chosen to rewrite the hypothesis one completely

Point 35: 205-206 High school spans differ between countries, rather provide the age of participants (range) 

Response: We have modified the first part of the section “Sample”

Point 36: 218- 219 “characteristic, a psychosocial questionnaire was administered to all the participants” 

Response: We have rewritten the sentence

Point 37: 225 “ ... big five (OCEAN) model [44].” 

Response: We tried to better explain the I-TIPI test and its theoretical basis.

Point 38: 244-247 Merge in one paragraph. 

Response: Done.

Point 39: 248-251 Merge in one paragraph. 

Response: Done.

Point 40: 252-259 Make it the first paragraph of the section. 

Response: Done.

Point 41: 261 “This two phases experiment was made at school...” 

Response: Corrected.

Point 42: 261-262 Remove “was composed of two phases that”. 

Response: Done.

Point 43: 265 Use Figure instead of Fig. in your text 

Response: Done.

Point 44: 265-269 Merge in one paragraph 

Response: Done.

Point 45: 274 Replace “5 participants for” by “participants of”. 

Response: Done.

Point 46: 278 “least”, not “list”.  

Response: Done.

Point 47: Figure 4 Give again the definition of both groups in the caption, the state again the acronyms, and use acronyms in the graph. 

Response: We rewrote all the figures captions

Point 48: 316 The use of a multivariate model should have clearly appeared in the method section. 

Response: We improved the section analysis explaining both why and how we performed the GLMM.

Point 49: 323 “wide” instead of “width” 

Response: Done.

Point 50: 333-334 Remove “proposed here". 

Response: Done.

Point 51: Table 1 Could the table be formatted to have Factors described in one line only? 

Response: Done.

Point 52: 384 “the predisposition to form groups” Detail your thought cite the reference. 

Response: Done

Point 53: 359 “average members’ intelligence” really? Restate by rather referring to their RAPM scores. 

Response: Done.

Point 54: 386 “for a single individual”? for the average individual? 

Response: yes…

Point 55: 365-366 “First of all, ... performance decrease” Isn’t it obvious?

Response: actually .. we underlined it in the text

Point 56: 354 What do you mean by “the importance of the group”? Explain

Point 57: 383-393 Consider using these lines in the conclusion section.

Response: Since discussions are widely modified the following minor comment no fit anymore the manuscript.

Round 2

Reviewer 1 Report

Dear author, the document has improved significantly, thanks for attending to the improvement proposals.